# Family Self-Care Pattern in Families with Children with Intellectual Disabilities: A Pilot Study

**DOI:** 10.3390/healthcare13070791

**Published:** 2025-04-02

**Authors:** Teresa Dionísio Mestre, Manuel José Lopes, Ana Pedro Costa, Ermelinda Valente Caldeira

**Affiliations:** 1Health Department, Polytechnic Institute of Beja, 7800-111 Beja, Portugal; 2Comprehensive Health Research Center [CHRC], 7004-516 Évora, Portugal; mjl@uevora.pt (M.J.L.); anapedrocosta92@gmail.com (A.P.C.); ecaldeira@uevora.pt (E.V.C.); 3Health Department, University of Évora, 7000-811 Évora, Portugal; 4Local Health Unit of Lower Alentejo [ULSBA], 7801-849 Beja, Portugal

**Keywords:** family, family nursing, intellectual disability, nursing, self-care

## Abstract

Family self-care emphasizes a family’s role in health promotion and protection, reflecting society’s views on health, illness, and human relationships. In families with children with an intellectual disability, where the child may lack self-care abilities, family self-care becomes crucial, highlighting that self-care needs exceed individual capacity and require family cooperation. **Background/Objectives**: This pilot study aims to explore the factors influencing family self-care and define attributes of its cognitive, psychosocial, physical, and behavioral domains in families with children with intellectual disabilities. **Methods**: A descriptive and correlational study with forty-four families was conducted. Exploratory analysis and linear regression analysis were estimated through the assumptions of the Gauss–Markov theorem (specifically homoscedasticity, normality, and model specification adequacy). Multicollinearity was also evaluated. **Results**: The significant family conditioning factors identified were family income, education level, degree of physical and functional dependence of the child, family household size, and social support. Socioeconomic, demographic, and health-related factors shaped self-care experiences. **Conclusions**: Family empowerment and the impact of disability are key elements in enabling self-care. Families reporting a greater impact of their child’s condition tended to feel less empowered, directly affecting their ability to perform daily self-care activities. The evidence suggests a pattern in which self-care activities might be reactive rather than proactive and focused on managing immediate challenges rather than long-term well-being. These insights can guide healthcare professionals, especially family nurses, toward a holistic, family-centered approach to supporting families with children with intellectual disabilities.

## 1. Introduction

Changes in demographic trends, epidemiological patterns, and societal family structures have led to the emergence of new health demands [1]. In response, holistic care, particularly in nursing, has been widely promoted as a guiding philosophy that emphasizes not only person-centered but also family-centered care [2]. This approach aligns with the evolving healthcare paradigm, transitioning individuals from passive consumers to active participants in managing their health.

The increasing focus on self-care (SC) has encouraged family health nurses to extend its scope, directly linking it to family self-care (FSC). In this framework, the family is recognized as both a system and a social unit, where the whole exceeds the sum of its parts [3,4], with SC performing as a fundamental dimension [5]. Today, SC encompasses multiple interpretations and dimensions, analyzed from various perspectives, reflecting its integration into healthcare discourse and the rising prevalence of chronic diseases [6]. By fostering SC behaviors, families can improve their overall well-being, effectively managing health conditions and promoting long-term health [7].

The World Health Organization [5] recently redefined SC as the capability of individuals, families, and communities to actively promote health, prevent illness, and sustain well-being, even in the presence of diseases or functional impairments. This definition emphasizes that SC can be developed independently or with the assistance of healthcare professionals. Within this framework, SC is regarded as a crucial health resource that significantly contributes to well-being and quality of life in a family setting. While SC plays a fundamental role at the individual level [8,9], the family serves as the primary environment where it is cultivated [10]. However, rather than being a mere sum of individual SC practices, it manifests as an FSC pattern, which becomes particularly relevant at key transitional moments in the family life cycle [4]. This pattern is intrinsic to the family unit, shaped by each member’s contributions. However, its full significance emerges when the family is understood as an interconnected system and a social unit [4].

As families develop, they inevitably experience transformations that bring about change. Understanding the effects and significance of these changes is crucial, as they may stem from critical events or disruptions that alter family dynamics, including ideals, perceptions, identities, relationships, and daily routines [11].

For families raising children with intellectual disabilities (IDs), where the child lacks—or will not develop—the ability to independently engage in SC, the FSC pattern becomes particularly vital. It plays a fundamental role in sustaining life, health, and well-being, reinforcing the notion that SC often extends beyond individual capacity and necessitates collective effort [8]. In this context, FSC is practiced within family, community, and home settings, emphasizing the interconnected and cooperative nature of caregiving [10].

The recognition of an ID in a child represents a significant life event that can profoundly impact parents, often disrupting multiple aspects of daily life [12]. In particular, it has been shown to negatively affect the psychological well-being and overall mental health of the family unit [13]. Research highlights that families raising a child with an ID frequently experience elevated levels of stress, including financial strain due to high medical expenses [14], heightened anxiety [15], depression [16,17], and reduced psychological well-being [13], especially in the absence of effective coping strategies and adequate social support [18,19,20]. Beyond these psychological and financial challenges, caring for a child with an ID can also reduce caregivers’ confidence and perceived competence in their parenting abilities [21]. Given that family members are deeply interconnected, any challenge affecting one individual inevitably influences the entire family system [22]. Recognizing the central role families play in providing care for children with health conditions is essential, particularly in community-based settings, where increased support and acknowledgment are needed to strengthen their caregiving capacity.

Intellectual disability is classified as a subgroup within neurodevelopmental disorders [12], characterized by impairments in neurodevelopmental functioning and marked by significant limitations in both intellectual and behavioral abilities [23]. The evaluation of intellectual functioning in individuals with IDs focuses on cognitive processes such as problem-solving and reasoning, encompassing conceptual, practical, and social domains, including SC, social interaction, and learning skills [24]. The etiology of ID is diverse, including prenatal factors such as genetic disorders, metabolic conditions, cerebral malformations, and maternal illnesses. Environmental influences, including intrauterine exposure to alcohol (responsible for approximately 8% of mild cases), toxins, and teratogens, are also recognized contributors. Additionally, neonatal and postnatal factors, such as intrapartum asphyxia, hypoxic–ischemic injuries, traumatic brain injuries, infections, demyelinating diseases, seizures, and metabolic disorders, have been implicated in ID development [23,25,26]. Among cases with an identifiable cause, trisomy 21 (Down syndrome) remains the most prevalent genetic contributor, whereas fragile X syndrome represents the leading hereditary cause of ID [27,28]. ID is often associated with a range of neuropsychiatric and neurobehavioral conditions, including autism spectrum disorder and epilepsy, as well as neuromuscular impairments such as ataxia, spastic paraplegia, sensory or motor neuropathy, and muscular dystrophy [25,29,30].

The Diagnostic and Statistical Manual of Mental Disorders, 5th Edition (DSM-5), published by the American Psychiatric Association [23], defines an ID as a condition that originates during the developmental period and affects three key domains of daily functioning. These include (i) the conceptual domain, which encompasses knowledge, reasoning, memory, and literacy skills; (ii) the social domain, which pertains to communication abilities, maintaining relationships, and demonstrating empathy; and (iii) the practical domain, which involves self-care, managing daily responsibilities, attending school, maintaining employment, and handling finances [23]. DSM-5 classifies ID into mild, moderate, severe, and profound categories, based on levels of adaptive functioning rather than IQ alone [23].

The level of support and healthcare needs of a child with ID differs depending on the severity and nature of the condition [17]. However, due to their limited independence, children with IDs remain highly dependent on caregivers, often requiring lifelong support from their families [22,31,32,33]. The ongoing caregiving responsibilities significantly impact parents and caregivers, contributing to higher levels of distress and an increased need for external support systems compared to families of neurotypical children [31]. This burden manifests in multiple areas of parental life, driving families to seek assistance from both informal and formal support networks. In addition to the emotional and financial strain, caregiving for a child with an ID demands constant vigilance, adherence to medical routines, and frequent healthcare appointments, placing substantial time constraints on families [34]. Given that these children require specialized care beyond what is typically needed for neurotypical development [17], early professional intervention is essential [13]. Healthcare professionals play a pivotal role in supporting families, providing crucial information, and equipping caregivers with the necessary skills to manage their child’s condition effectively [35].

Healthcare professionals play a crucial role in promoting the well-being and development of families and children with IDs [36] by adopting FSC as a structured care model. Within this framework, if families are primarily responsible for meeting the health needs of their members, it is essential that they have access to the necessary resources, opportunities, and well-being to fulfill this role effectively [10,37]. The FSC pattern becomes particularly significant at specific stages of the family life cycle, particularly when caring for children with IDs who, due to their limitations, are unable to engage in self-care independently [4].

Despite its importance, FSC remains underexplored in both research and practice, as existing health and social care systems have not systematically incorporated it into structured models of care. This gap in evidence highlights the need for further theoretical, empirical, and applied research to support its development [4]. Currently, there is a lack of scientific consensus on the absolute necessity of FSC and the most effective ways to integrate it into real-world family settings [38]. Recent studies have begun to address this gap by positioning FSC within the context of three theoretical frameworks: Orem’s Self-Care Nursing Theory, Riegel’s Middle-Range Theory of Self-Care in Chronic Disease, and von Bertalanffy’s General Systems Theory [4]. These theories conceptualize the family as a dynamic system and social entity, reinforcing the interconnection of SC and family structure. A recent study [4] established independence, responsibility, and autonomy as core principles of FSC, emphasizing its reciprocal relationship with the community and surrounding environment. To ensure effective maintenance, monitoring, and self-management of family members with chronic illnesses or disabilities, four key domains—cognitive, physical, psychosocial, and behavioral—have been identified as fundamental to FSC development [4].

Given the scarcity of research on FSC in families with children with IDs, this study serves as a pilot to explore key conditioning factors and test the feasibility of methodologies for future large-scale studies.

From this standpoint, the present study aimed to identify the conditioning factors that influence the development of FSC and to define the attributes associated with its cognitive, psychosocial, behavioral, and physical domains. Existing literature on this topic remains scarce; however, key conditioning factors identified include the age and gender of family members, educational background, sociocultural influences, overall family health status, the presence of pathological conditions within the family, and the stage of the family life cycle [10,39]. In this study, we considered the family’s needs as a system and social entity, recognizing SC as a foundational standard of care that directly impacts family health. These considerations underscore the need for further research to clarify the structure, development, and practical application of FSC. A deeper understanding of this care pattern is essential for advancing family nursing practices and enhancing the support provided to families navigating complex health challenges.

## 2. Materials and Methods

### 2.1. Design

A pilot descriptive and correlational study design [40] was selected to describe the family’s sociodemographic data, the clinical data of the children with ID, the impact of a child with deficits/chronic disease on the included families, their perceptions of community support, their satisfaction with care, and the interrelationships between these factors through the evaluation of family empowerment. The data were collected between March and December 2023.

As a pilot study, this research involved a small, non-randomized sample (N = 44) to test the feasibility of data collection methods and assess preliminary trends. The findings will inform the design of a larger, more representative study.

### 2.2. Setting

This study was conducted in Portugal, more precisely, in Lower Alentejo, in an area covered by the Local Health Unit of Lower Alentejo. All families with children with previously diagnosed IDs were identified by family nurses at the Primary Health Care Service of 13 Local Health Centers, with one nurse acting as a bond to the research team (mostly the head nurse of each health center). It took place in community settings, and family nurses made the first contact with the children’s families (parents and/or caregivers). This first approach was taken by family nurses (due to their proximity) to investigate families’ willingness and interest in taking part in the study. The address and telephone number of the parents/caregivers were subsequently provided to the lead researcher. The data collection instruments were applied on paper and in person at the place where the families felt most comfortable (home, health center, or workplace). This approach was intended to meet the intrinsic context of each family, understood as a place to facilitate the data collection process. Additionally, there was space for parents to speak about their daily routines and to clarify any doubts about filling in the instruments. These meetings were held by the lead researcher.

### 2.3. Instruments

When we have a phenomenon that is still poorly understood, as is the case here, researchers first try to gather as much information as possible about it to identify the many areas of the phenomenon [40]. The following four items refer to the different data collection instruments used in this study: the first is an instrument used to collect sociodemographic data from the families under study; the following three correspond to scales validated and translated for the Portuguese population. They were applied to parents or caregivers of children with IDs. These instruments aimed to describe the characteristics and development of the FSC domains identified in the literature [4].

#### 2.3.1. Sociodemographic Data

Sociodemographic data comprised the data collected from the members of each family, including data from the parents/caregiver (age, gender, relationship, family situation [marital status], religion, household composition, level of education, professional situation, source of income, monthly average, contact made and/or maintained with health services [frequency, duration, professionals involved], conditions/characteristics of residence, and resources in the community and from the child (age, gender, care provided on a daily basis), as well as a description of clinical data about the diagnosis (diagnostic category; time of diagnosis). The collection of these data made it possible to identify the factors that may affect the development of FSC.

#### 2.3.2. Barthel Index (BI)

One of the instruments used to assess the functional capacity to carry out basic activities of daily living (BADLs) is the Barthel Index (BI), designed by Mahoney and Barthel [41]. The psychometric qualities of this instrument are documented in studies with elderly patients and/or those with stroke, although some researchers have used it with people with other diagnoses [42]. The BI is thus an instrument that assesses a person’s level of independence in performing ten BADLs [41].

The BI has been validated and translated into Portuguese by Araújo et al. [43]. Despite its extensive applicability in hospital settings, a study on the fidelity of scales in the community confirmed that it is an instrument with excellent fidelity [43]. In this case, it was filled in by the parents/caregivers, considering their child’s deficits. In this specific family context, this scale aims to assess the child’s physical and motor deficits, which the family ensures daily.

#### 2.3.3. Impact on Family Scale (IOFS)

To study the consequences of a child’s chronic illness on the family system, Stein and Riessman developed a measure of family impact in 1980 for use in a clinical study of children with heterogeneous diagnoses: the IOFS [44]. The Portuguese version of this scale was translated and validated by Albuquerque et al. [45]. Although the original version of this scale was specifically designed to assess the impact of chronic illness, its subsequent application has been extended to other areas, including disability [46]. This instrument is intended to assess parental perceptions of the effects of the child’s health condition on family life and can be used with different diagnostic groups. In the Portuguese version, the scale consists of 15 items, answered on a four-point Likert scale indicating the degree to which each statement applies to the parent or family. This scale enables the recognition and identification of characteristics inherent in the cognitive and psychosocial domains, facilitating the development of FSC and the achievement of the family’s [self-]management of disability.

#### 2.3.4. Family Empowerment Scale (FES)

The FES was developed by researchers at the Research and Training Centre on Family Support and Children’s Mental Health at the University of Portland to assess family empowerment in the context of mental health services for children [47]. In this instrument, states of empowerment are dynamic, changing over time depending on experiences, new conditions, or circumstances involving the family. The structure of the FES covers two dimensions: the level of empowerment and the way empowerment is expressed.

In Portugal, the scale was translated and validated by Melo and Ferreira [48]. It consists of 34 items, and each item can be scored on a five-point Likert scale. It was completed by parents or caregivers who were involved in the daily care and health maintenance of the child with an ID. The score assessed by the parents/caregivers represents the family empowerment of the family unit, with higher scores indicating greater family empowerment. Through the two dimensions elucidated, this instrument made possible the transposition of the four FSC domains and their associated characteristics. If the family feels empowered, it will be able to develop SC behaviors more effectively [4].

### 2.4. Ethical Considerations

This study received ethical clearance to begin with the empirical data collection process from the Ethics Committee of the University of Évora (reference number UE_22083, November of 2022) and the Ethics Committee of the Local Health Unit of Lower Alentejo (reference number ULSBA/01/2023, January of 2023), Portugal. Approval was given in accordance with the International Ethical Guidelines for Health-related Research Involving Humans. Authorization to use the data collection instruments was requested and granted to the authors, who validated them for the Portuguese population.

Participation in this study was voluntary. In the first contact, primary caregivers were assured in written informed consent that their data would remain anonymous and that their participation would not affect their current health services or daily life. The purpose of the study was explained, and parents were assured and always protected with respect to all issues regarding confidentiality and privacy.

### 2.5. Participants

The eligibility criteria for participation included family members aged 18 or older who provided direct care to children with IDs in a home setting, demonstrated cognitive ability to complete the data collection instruments, and willingly consented to participate in the study. Additionally, based on the researcher’s judgment, families were included if they had children aged over 2 years with a confirmed ID diagnosis established by healthcare professionals, ensuring the study focused on cases where the condition was formally recognized.

The study aimed to encompass the entire target population via a non-probabilistic convenience sampling method. Although 54 families were identified, only 44 agreed to participate. All the data collection instruments were applied to the same person(s), typically the parents or primary caregivers.

### 2.6. Data Analysis

All the instruments used were anonymous but were coded to ensure that the answers could be cross-referenced. The instruments described were included in the analysis, and they were excluded if 10% or more of the items were not completed. For the exploratory analysis, means, standard deviations, and total scores were calculated for each quantitative variable, and qualitative variables were analyzed on the basis of the respective absolute and relative frequencies. No outliers were detected for any of the variables. The exploration analysis was conducted via Microsoft Excel to Microsoft 365 MSO (version 2502 Build 16. 0. 18526. 20168) 64-bit and R in the RStudio (R 4.2.0) environment. Additionally, Excel was used to create graphical representations of data, including histograms and circular charts, to visually summarize the distribution and trends within the dataset.

The Gauss–Markov theorem was applied in this study because Ordinary Least Squares (OLS) regression was used to examine the relationships between FSC practices and key predictor variables, such as family empowerment or the impact of a child with ID on the family. According to the Gauss–Markov theorem, in a linear regression model where the assumptions of linearity, independence, homoscedasticity, and imperfect multicollinearity are met, OLS estimators are the best linear unbiased estimators. This means that OLS provided the most efficient (i.e., minimum variance) estimated among all unbiased linear estimators [49].

Firstly, we explored the relationship between the FES and IOFS. Specifically, simple linear regression was used to assess how FES collectively predicts IOFS while controlling for potential confounding factors. This model, Equation (1), was estimated via the OLS method, and the assumptions of the Gauss‒Markov theorem regarding error behavior were tested, specifically homoscedasticity (Breusch‒Pagan test), normality (Jarque‒Bera test), and model specification adequacy (RESET test) [49].

Model 1.(1)$$\;{\;FES}_{i}=\beta_{0}+\beta_{1}{IOFS}_{i}+\varepsilon_{i}$$

Model 2 was subsequently estimated via the same estimation procedure, with the FES as the dependent variable and the most significant variables identified in the literature, namely, age, sex, education level, income, religion, family household, pathological disorders of family members, support networks, and health resources [10] as independent variables. Multicollinearity was also evaluated via the variance inflation factor (VIF) method [49].

Model 2.(2)$$\begin{array}{cc} {FES}_{i}= & \beta_{0}+\beta_{1}{Barthel}_{i}+\beta_{2}{Sex}_{i}+\beta_{3}{Age}_{i}+\beta_{4}{Income}_{i}+\beta_{5}{Religion}_{i}+\beta_{6}{Educ}_{i}\\ & \;\;\;\;+\beta_{7}{Family}_{i}+\beta_{8}{Health\_inst}_{i}+\beta_{9}{Distance}_{i}+\beta_{10}{Support\_family}_{i}\\ & \;\;\;\;+\beta_{11}{Support\_friends}_{i}+\beta_{12}{Support\_neighbours}_{i}\\ & \;\;\;\;+\beta_{13}{Social\_support}_{i}+\varepsilon_{i} \end{array}$$

Assuming that IOFS was the dependent variable, Model 3 was also estimated through linear regression analysis, as was the case with Model 2.

Model 3.(3)$$\begin{array}{cc} {IOFS}_{i}= & \beta_{0}+\beta_{1}{Barthel}_{i}+\beta_{2}{Sex}_{i}+\beta_{3}{Age}_{i}+\beta_{4}{Income}_{i}+\beta_{5}{Religion}_{i}+\beta_{6}{Educ}_{i}\\ & \;\;\;\;+\beta_{7}{Family}_{i}+\beta_{8}{Health\_inst}_{i}+\beta_{9}{Distance}_{i}+\beta_{10}{Support\_family}_{i}\\ & \;\;\;\;+\beta_{11}{Support\_friends}_{i}+\beta_{12}{Support\_neighbours}_{i}\\ & \;\;\;\;+\beta_{13}{Social\_support}_{i}+\varepsilon_{i} \end{array}$$

The estimation of all the models and the verification of the Gauss‒Markov theorem assumption were performed via R in RStudio software (R 4.2.0).

Given our focus on analyzing the relationship between FSC and key influencing factors, we sourced descriptive statistics to summarize sample characteristics (e.g., means, standard deviations, and frequencies). This necessary analysis provided context before conducting inferential analyses. The correlation analysis through Pearson’s correlation was used to explore associations between variables such as family empowerment (FES) and the impact on the family (IOFS). These allow the identification of potential predictor variables for regression analysis. With the regression analysis (OLS), we intended to determine the magnitude and direction of the relationship between FSC and influencing factors, selected due to its robustness in analyzing continuous dependent variables. The Gauss–Markov theorem justified the use of OLS, ensuring the efficiency and reliability of estimates. The diagnostic tests confirmed that OLS was appropriate and that the assumptions for valid inference were met.

## 3. Results

The study involved a total of 44 families, representing 81.48% of the target population, ensuring a representative sample. To simplify the presentation and understanding of the data, this section is divided into five subsections.

### 3.1. Characteristics of the Participants, Their Children with ID, and the Family Unit

The majority of participants were female (97.7%), specifically 42 mothers, 1 grandmother, and 1 father. The mean (SD) age of the participants was 39.3 (±7.8) years (Figure 1). All participants were of Portuguese nationality, although five had dual nationality (n = 3 Brazilian, n = 1 Moldovan, and n = 1 Ukrainian).

With respect to participant characteristics, 37 (84.09%) were married or in a civil partnership, 4 were divorced, and 3 were single. According to religion, 20 families reported that they practice a religion, with 18 identifying as Catholic and 2 as evangelical. The family households ranged from two to five people, with the most common configuration being families consisting of a mother, father, and two children (n = 24). Importantly, 12 families had only one child (the one diagnosed with ID).

In terms of employment status, 32 respondents were employed full-time, 10 were unemployed, 1 was receiving social inclusion income, and 1 was retired. All unemployed individuals were full-time mothers/housewives. Furthermore, 31.8% (n = 14) had graduated from high school (the current mandatory education level in Portugal), and 34.1% (n = 15) had achieved bachelor’s or master’s degrees. The remaining 34.1% (n = 15) had education levels below high school. The monthly family income ranged from less than EUR 600 to more than EUR 2000. Given that the minimum salary in Portugal is currently EUR 820, 14 (32%) have incomes of up to EUR 1000 a month, 18 (41%) have incomes between EUR 1000 and EUR 2000, and 12 (27%) have incomes over EUR 2000 a month.

Regarding the place of residence, 79.5% (n = 35) lived in towns, villages, or farms spread throughout Lower Alentejo, and only 20.5% (n = 9) lived in cities.

In terms of the families’ support networks and health resources, 32 (72.7%) identified the family as the main support network for the child with ID, followed by friends (n = 8, 18.2%), neighbors (n = 2, 4.55%), and a full-time nanny (n = 2, 4.55%). The most common health resources available were health centers (primary health care), identified by 38 participants (86.4%). The average distance from the home environment to the closest health service was 10.01 km, ranging from 0 to 55 km.

In terms of contact and proximity to health professionals, 41 families (93.2%) reported greater contact with doctors, 31 (70.5%) with nurses, 30 (68.2%) with therapists (physiotherapists, occupational therapists, speech therapists, and psychomotor therapists), and 18 with psychologists. Many respondents chose more than one of these professional categories. Due to proximity, therapists (n = 26) and nurses (n = 23) were the most common.

Most of the children with IDs were male (n = 29), representing 66% of the sample and the remaining 34% were female (n = 15). The age range of the sample was 3–15 years, and the age at diagnosis ranged from birth to 8 years. Our sample included 11 different diagnoses related to IDs: 16 cases of autism spectrum disorders, 10 cases of Down syndrome, 5 cases of cerebral palsy, 4 cases of genetic/metabolic syndrome under study, 2 cases of microcephaly, 2 cases of fragile X syndrome, and 1 case each of Syngap 1 syndrome, Turner syndrome, Lesch‒Nyhan syndrome, chromosome 2 inversion, and neuronal ceroid lipofuscinosis (NCL5). Most of these children require and take medication daily (n = 32). On the basis of the diagnosed conditions and respondents’ answers, 22 children were mostly dependent on BADLs (considering the stage of development, they were asked to identify the functional deficits resulting from the child’s diagnosis). The remaining children were independent in at least one basic activity of daily living. All the family units ensured the daily SC activities of their children with IDs at home.

### 3.2. Barthel Index (BI) Results

The BI was completed by the parents or caregivers, indicating complete dependence for 10 children scoring less than 20 points and severe dependence for 4 children scoring between 20 and 35 points. There were 8 children with moderate dependence and 22 with mild dependence, and no children were independent in all BADLs. The mean (SD) BI score was 47.5 (±31.5), with a minimum of 0 points and a maximum of 85 points. Importantly, the mode was 0 points, highlighting complete dependence as dominant in the BADLs of the children with ID.

### 3.3. Impact on Family Scale (IOFS) Results

The IOFS highlights parents’ or caregivers’ perceptions of the burden associated with caring for a child with an ID, revealing their cognitive and psychosocial competences. The mean (SD) IOFS score was 36.11 (±7.7), with a minimum of 18 points and a maximum of 51 points. This scale indicates a greater perception of the impact of the family’s health condition for scores between 45 and 60 points, and seven families in our sample fall within this range.

### 3.4. Family Empowerment Scale (FES) Results

Since this scale can range between 34 and 170 points, our sample scored between a minimum of 84 points and a maximum of 165 points. The mean (SD) FES score was 128.57 (±19.96), and the mode score was 158 points. Considering scores above 140 points as a high level of family empowerment, we had 15 families in this category. For moderate family empowerment levels, 26 families had scores between 100 and 140 points.

With respect to the three levels of family empowerment assessed by the FES, i.e., family, care provided to the child, and community involvement, it was possible to evaluate each one separately. At the family level (involving the first 12 items), an average (SD) score of 47.86 (±6.97) was obtained, with a maximum of 60 points and a minimum of 34 points. At the level of care provided to the child (also with 12 items), a mean (SD) score of 49.38 (±7.83) was obtained, with a maximum of 60 points and a minimum of 37 points. At the community involvement level (the last 10 items), a mean (SD) score of 31.31 (±7.45) was obtained, with a maximum of 48 points and a minimum of 15 points. These data indicated that the level of care provided to the child demonstrated the highest level of family empowerment, followed closely by family management in everyday situations.

### 3.5. Factors Associated with Family Self-Care Development

A regression analysis was performed relating the FES score to the IOFS score, with the FES score considered the dependent variable (Table 1). On the basis of the results presented in Table 1, the variables are negatively correlated, which means that when the IOFS increases by 1 unit, the FES tends to decrease by 1.027 units. Overall, the model is statistically significant, although the coefficient of determination is low at 15.5%. This finding indicates that when IOFS increases, FES decreases, meaning that the greater the impact of the ID associated with the child and felt by the family, the lower their level of family empowerment.

Following the linear regression analysis with FES as the dependent variable for the development of FSC, Model 2 was estimated (Table 2). The results showed that the assumptions of normality and homoscedasticity of the error are met, and both models do not reject the null hypothesis of the RESET test, which postulates that the model specification is correct. Since the overall model is not statistically significant (F statistic = 1.265), it was decided to estimate a more restricted model with the variables that showed statistical significance. Given the importance of the BI variable, it was also included in the restricted model. This restricted model was statistically significant (F statistic = 2.707), with only two variables showing statistical significance: income and education level.

Income was significant in a negative direction, which means that the lower the household income was, the greater the level of family empowerment. Education level had the opposite sign and was significant in a positive direction. This means that the higher the participants’ level of education is, the greater their level of family empowerment. The results of the multicollinearity analysis (Table 3) indicated no evidence of significant multicollinearity between the independent variables.

Model 3 was estimated considering the IOFS score as the dependent variable (Table 4). The results showed that the assumptions of normality and homoscedasticity of the error are met, and both models do not reject the null hypothesis of the RESET test, which postulates that the model specification is correct. In Model 3, the variables Barthel (negative sign), income (positive sign), education level (negative sign), and family (negative sign) are statistically significant. Since the overall model is not statistically significant (F statistic = 1.85), a more restricted model with the variables that showed statistical significance, including those referring to support obtained from friends and social support, was used. This inclusion is justified by the multicollinearity between these variables and the variable referring to Family_support. This restricted model is statistically significant (F statistic = 3.98), showing statistical significance for the following variables: Barthel, income, education level, family household, and social support (Table 4).

In this analysis, a lower BI score was associated with a greater impact of a child with an ID on the family. Income presented the opposite effect, indicating that families with higher incomes experience a greater impact on their family’s health condition. The level of education revealed that participants with lower levels of education perceived a greater impact on the family. Additionally, smaller households reported a greater perceived burden associated with caring for a child with ID. Social_support became significant (negative sign) in the presence of the variable Support_friends, indicating that a larger social support network reduces the impact of a child with an ID on the family. The results of the multicollinearity analysis (Table 5) indicated no evidence of significant multicollinearity between the independent variables in Model 3.

## 4. Discussion

The current study is the first to report on the FSC pattern in a sample of families with children with ID in Portugal. The main goal is to identify and explain the factors influencing its development and define the attributes of the four domains identified through the literature. The findings provide initial insights into the FSC pattern. However, given the pilot nature of this study, further research with larger samples is necessary to confirm these trends. Our understanding of how these families with children with ID develop FSC is primarily based on conceptual findings from two studies that consider the family as a system and social unit, highlighting their current SC practices [4,10].

This study reveals a complex interplay of socioeconomic, psychosocial, and health-related factors that shape these families’ SC experiences and coping strategies. The statistically significant conditioning factors include family income, the education level of the parents/caregivers, the degree of physical and functional dependence of the child with an ID, the size of the family household, and the social support network. In this context, the family’s stage of development and the specific ID condition associated with the child influence the family unit (although indirectly), contributing to the factors listed above.

Through the characteristics of the participants, their child with an ID, and the family unit, 43 of the 44 families studied are in the stages of the life cycle of families with young children and adolescents. In this context, the family assumes a set of tasks and functions aimed at ensuring the survival, development and well-being of these children [31,50]. In the exercise of the parenting role, they assume support, protection, affection, and love, among other tasks and functions, performed throughout the life of children, according to their personal characteristics and the environmental factors that surround them, providing conditions for the protection, conservation and promotion of health [51,52]. Traditional or nuclear families were the most common in this study (n = 24), and females were the major respondents to the instruments (97.7%). In accordance with these results, we can see gender disparity, with the maternal figure standing out as the primary caregiver and assuming a prominent position in the family unit, even as a representative for participating in this study. This establishes a potential challenge to the FSC pattern, as the data were obtained almost exclusively from a single family member, despite attempts to capture the entire family’s perspective.

The study was conducted in southern Portugal, a region known for its rich cultural heritage but facing several sociodemographic challenges, including resource scarcity and economic disparities [53,54]. A significant percentage of the population in Lower Alentejo lives below the national income average [53,55], and many areas lack adequate health facilities, forcing residents to travel long distances to receive care [56]. Our study highlights these barriers, with the majority of the families studied (n = 34) having incomes ranging from less than EUR 1000 to EUR 2000, supporting an average of four people per household.

With respect to the FSC domains, it is important to note that they are defined by the characteristics and particularities of the families studied. Family empowerment significantly contributes to enabling SC within the family. A key finding of our study is the inverse relationship between family empowerment (FES) and the impact on the family (IOFS) when a disability is present in one of the family members (β1 = −1.027, *p* < 0.01). The findings suggest that family empowerment is a central determinant of FSC practices. Empowered families are more likely to navigate healthcare systems effectively, advocate for their child’s needs, and implement SC strategies that improve overall well-being. These results, aligned with previous research, indicate that caregiver confidence, knowledge, and autonomy directly affect health outcomes in children with ID. Some studies [10,14,37] also emphasize that family support networks and community resources strengthen empowerment, mitigating the negative impact of caregiving burden.

Family empowerment, being a crucial factor in FSC, reflects not only how the family manages SC activities daily but also their involvement with the surrounding community and environment. This empowerment reflects the family’s attitudes, knowledge, and behaviors toward their health situation and condition. Additionally, the impact that a deficit or disease can have on the family is significant for their psychosocial and cognitive abilities to manage SC [4]. Families reporting a greater impact on their child’s condition tended to feel less empowered. These results corroborate the findings of Bujnowska et al. [20] and Lee and Burke [57], who suggest that the challenges associated with caring for a child with ID can significantly affect the family’s sense of control and efficacy. From this perspective, healthcare professionals, particularly nurses, play a critical role in fostering family empowerment [13,36], helping caregivers transition from reactive responses to proactive SC strategies, thereby promoting resilience and long-term well-being. Targeted interventions, such as structured follow-up consultations, educational workshops, and access to peer support groups, can enhance caregivers’ skills and confidence in managing their child’s care.

As no studies comparing the results of the applied models were found, we will now conduct an individualized assessment of each FSC domain’s attributes, relating them to the statistically significant factors.

### 4.1. Cognitive Domain

Effective SC in this domain requires families to be well informed about their child’s diagnosis, treatment options, and daily care requirements. The study highlights the value of parental knowledge and self-management skills in ensuring family well-being. Higher education levels among parents are related to easier FSC development, suggesting that education plays a crucial role in enhancing the cognitive capabilities necessary for effective SC. This positive correlation between education level and family empowerment in Model 2 (β6 = 9.158) aligns closely with the literature. Higher education levels enhance problem-solving skills [58], improve access to resources, and increase health literacy [59], all of which contribute to a greater sense of empowerment. This finding suggests that educational interventions could be valuable tools for enhancing SC behaviors. However, the data also reveal that families with lower education levels face significant challenges, which may influence FSC development. Providing accessible and understandable health education materials, as well as training programs aimed at improving health literacy, can empower these families [60]. Simplified educational materials, such as visual guides, step-by-step instructional videos, and community-based workshops, can enhance health literacy and empower these families. Additionally, peer mentorship programs and healthcare navigation support can help them to better understand medical information and navigate healthcare services, ensuring equitable access to essential resources and fostering sustainable SC behaviors [60].

### 4.2. Psychosocial Domain

The psychosocial domain involves attitudes, values, desires, motivation, and perceived competence in carrying out SC actions. The study highlights the crucial role of social support (β13 = −4.7685) in reducing the impact of a child’s health condition on the family (IOFS as the dependent variable in Model 3).

Social networks provide emotional support, practical help, and a sense of community, which are essential for coping with the psychosocial stressors associated with raising a child with ID [61]. The presence and quality of social support networks are key conditioning factors in this domain [10]. For example, Skok et al. [62] reported that social support has a moderate role in mediating the impact of stress on the well-being of mothers of children with cerebral palsy. Lu et al. [63] noted that social support mediated the relationship between parental pressure and quality of life for parents of children with ID.

Highlighting this evidence, family nurses can facilitate access to community-based support networks, peer mentoring programs, and structured caregiver support groups [60,62]. These initiatives could help families build resilience by providing emotional support, shared caregiving strategies, and guidance from experienced caregivers. Additionally, nurses can link families to institutional resources, such as care services, that allow caregivers to take necessary breaks and prevent burnout [61].

In this domain, and contrary to our initial expectations, we found that lower income was associated with greater family empowerment, as seen through the negative correlation in Model 2 (β4 = −6.947). Given that income is a conditioning factor for FSC, this may reflect greater resilience among lower-income families; more effective community support systems (with a preference for public hospitals and health centers), especially in rural areas (which constitute 79.5% of our sample); or even differences in expectations and definitions of empowerment across income levels. Conversely, higher-income families may face unique stressors or have higher expectations that influence their perceptions, resulting in a greater impact on the family’s health condition. These results suggest that families’ healthcare utilization shows a particular preference for public hospitals and health centers.

In Portugal, the National Health Service (NHS) is a publicly funded healthcare system that provides universal health coverage to all residents, ensuring access to necessary healthcare services for everyone. However, higher-income families may find the NHS insufficient in addressing the individual needs of their child with ID, reflecting the significant impact they feel. In contrast, lower-income families may view this healthcare system as an effective community health resource. A study by Irigoyen et al. [64] emphasized that families with higher incomes have more opportunities to seek the best benefits, such as health information or healthier environments, in the adoption of preventive health behaviors. This evidence suggests that the development of the psychosocial domain might vary across different socioeconomic levels.

### 4.3. Physical Domain

The physical domain addresses the family’s ability to manage SC actions related to the physical and functional capacities of all members. The findings indicate a spectrum of dependence, with families experiencing varying degrees of physical burden. Our analysis of the BI scores revealed that higher levels of child dependence (n = 14 with complete or severe dependence) were associated with a greater impact on the family. Model 3 shows that lower Barthel scores are significantly associated with IOFS (β1 = −2.5889). This underscores the intense challenges faced by families caring for these children. Constant care demands affect many aspects of family life [65], including financial burdens [66], social interactions, and parental health.

As a result, SC heavily focuses on meeting the child’s needs, emphasizing the assignment of well-defined tasks to all family members. This finding underscores the necessity for targeted support networks tailored to families with highly dependent children, providing individualized care and specialized training in caregiving procedures. In this regard, evidence shows that the engagement between health professionals and families is the key factor in developing childcare plans through effective partnerships, which are based on the development of communication skills and adaptive strategies [60].

Larger social support networks were associated with reduced family impact. Model 3 shows that greater social support reduces IOFS (β13 = −4.7685). These results suggest that social support plays a crucial role in these families and is a significant conditioning factor for FSC development. We emphasize the importance of both formal and informal support networks in mitigating the challenges faced by these families. Interestingly, smaller households reported a greater perceived burden, possibly because fewer family members were available to share responsibilities and caregiving. These findings also highlight the importance of engaging the community, society, and environment as essential resources.

In smaller households, where caregiving responsibilities are concentrated on one or two individuals, family nurses can assist in developing structured care routines, introducing assistive technologies, and facilitating respite care to alleviate the caregiver burden [67]. In larger households, these nurses can implement family-centered interventions that delegate caregiving roles among multiple family members, fostering shared responsibility and reducing emotional strain [67].

Notably, families with higher incomes reported a greater impact on their health condition, indicating that financial resources alone do not alleviate physical burdens. Income significantly influences the physical domain, yet higher income levels do not necessarily reduce physical burdens. This could be because families with higher incomes often have higher standards of care and expectations [68]. Interventions could address this by providing additional support services such as physical therapy and occupational therapy, which help manage the physical demands of caregiving. According to Black et al. [69], having access to necessary medical equipment and home modifications can help families alleviate physical challenges.

### 4.4. Behavioural Domain

This domain focuses on the skills needed to develop SC behaviors. Model 3 shows that as the household size (variable Family_household) increases, the IOFS decreases (β7 = −2.8185). It can also be understood that smaller family households correlate with a greater perceived burden, possibly due to limited support within the home. Household size and structure are critical factors influencing the behavioral aspects of FSC. Behavioral interventions aimed at developing SC skills are essential to alleviate this burden. Encouraging families to engage in activities that promote resilience and adaptive behaviors can further enhance FSC development. For example, workshops on problem-solving and stress management techniques can equip families with tools to better manage daily challenges. Also, integrating digital tools, such as telehealth consultations and mobile apps for SC tracking, can further enhance caregivers’ ability to manage their child’s needs proactively [60,70].

Interaction with the community and family environment is a recurring theme across all domains. We emphasize that the development of FSC is significantly influenced by family integration within their community. Families that actively engage with community resources [4,10], such as healthcare services, educational institutions, and social support networks, are better positioned to establish effective FSC patterns. The quality and accessibility of these community resources act as critical factors in fostering a supportive environment for FSC development and the achievement of family health.

Possibly, scheduled follow-ups and home visits allow family nurses to assess the evolving needs of caregivers, reinforce proactive SC strategies, and adjust interventions accordingly. By engaging in home-based assessments, nurses can identify environmental barriers to effective caregiving and provide tailored recommendations to optimize the home setting for SC management [67].

### 4.5. Limitations

This study has some limitations. As such, the results must be interpreted cautiously.

As this study was designed as a pilot, the small sample size and specific regional focus limit the generalizability. Future studies should include samples from multiple regions with diverse economic, social, and healthcare conditions. Additionally, cross-country comparisons could provide further insights into how different health systems and policies influence FSC patterns. Also, future research should involve larger, randomized samples to validate these findings (e.g., >100 families), enhance the robustness of statistical analyses, and validate the initial findings. Additionally, further studies could employ longitudinal designs or multi-center approaches to improve generalizability.

Another limitation was that the predominance of mothers (42 of the 44 participants) as respondents in our sample limits the scope of perspectives captured in this study. We recognize that this reliance on maternal reports may not fully capture the complexity of FSC dynamics, as fathers, siblings, and grandparents also play significant roles in caregiving. To address this limitation, future research should aim to increase participation from multiple family members to provide a more holistic view of FSC. This could include encouraging fathers or other caregivers to complete the questionnaires alongside mothers.

For future research, a more randomized or stratified sampling approach could enhance representativeness. Expanding recruitment to other healthcare services, social organizations, and online platforms could help reach a more diverse group of caregivers. Moreover, incorporating mixed-methods designs with qualitative interviews from different family members may help mitigate potential biases and offer a more comprehensive understanding of FSC patterns.

Similarly, the explanatory capacity of the models used in this study was low to moderate, even after a significant number of potentially explanatory variables were included. This result demonstrates that there may be many subjective factors in this process that cannot be quantified or explained via a quantitative approach. Additionally, the wide range of diagnoses represented in our sample (11 different types) underscores the heterogeneity of ID and the need for individualized approaches to family support. Future research could refine the approach by using stratified sampling based on levels of functional independence or by conducting comparative subgroup analyses to explore variations in FSC strategies.

## 5. Conclusions

This study provides valuable insights that contribute to understanding FSC in the context of families with children with IDs. By addressing the cognitive, psychosocial, physical, and behavioral domains and fostering constant interaction with the community, we can develop more effective strategies to support these families, ultimately enhancing well-being and quality of life in the acquisition and maintenance of SC. The family conditioning factors identified in this study—family income, education level, degree of physical and functional dependence of the child, family household size, and social support—serve as critical elements influencing the development of FSC and should be targeted in future interventions and policies. One notable implication of this study is the relationship between family empowerment and the impact of the child’s ID on the family unit. Families reporting a greater impact of their child’s condition tended to feel less empowered, which directly affects SC daily activities. This suggests a pattern where SC activities might be reactive rather than proactive and focused on managing immediate challenges rather than long-term well-being. Strengthening caregivers’ ability to make informed decisions and access healthcare resources can relieve caregiving burden and improve overall family well-being. Future interventions should prioritize family-centered approaches that promote autonomy, knowledge, and social support to enhance FSC outcomes in families with children with ID.

Moreover, expanding this research to diverse cultural contexts could provide a more comprehensive understanding of FSC. Comparative studies across different regions and populations would be particularly valuable in identifying the best practices.

Family nurses, policymakers, and healthcare institutions must collaborate to implement structured programs that enhance family’s ability to manage SC effectively.

Despite these elements providing insights into potential FSC patterns, it is crucial to acknowledge that a comprehensive understanding would require more specific data on actual SC practices and behaviors. Additional research focusing specifically on SC activities and routines might be needed within these families. While this study provides valuable insights into FSC from the perspective of primary caregivers, we acknowledge the need for a more inclusive and multi-perspective approach in future studies to enhance the validity and applicability of our findings.

## Figures and Tables

**Figure 1 healthcare-13-00791-f001:**
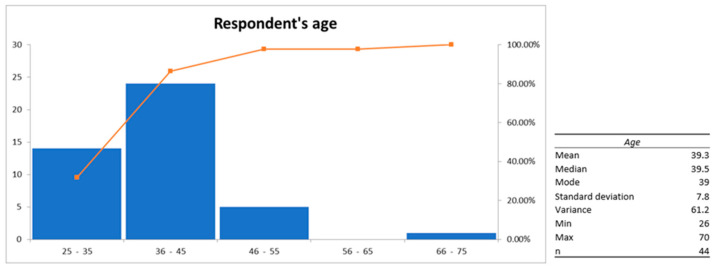
Descriptive analysis of the respondents’ age.

**Table 1 healthcare-13-00791-t001:** Results of the estimation of Model 1.

Dependent Var	FES	
(Intercept)	165.645	***
	(13.626)	
IOFS	−1.027	**
	(0.369)	
Observations	43	
F statistic	7.73	
R^2^	0.155	

Note: Estimation method OLS. The standard error of each parameter is in parentheses. The dependent variable is the FES. Significance codes: *** *p*< 0.001; ** *p*< 0.01.

**Table 2 healthcare-13-00791-t002:** Results of the estimation of Model 2 (Global and Restricted).

Dependent Var	FES			
	Model 2 Global		Model 2 Restricted	
(Intercept)	115.32		104.905	***
	(63.77)		(14.560)	
Barthel	2.369		1.620	
	(3.262)		(2.686)	
Sex	−14.03			
	(31.59)			
Age	0.085			
	(0.584)			
Income	−6.678	*	−6.947	**
	(3.212)		(2.720)	
Religion	5.291			
	(8.231)			
Educ_level	11.047	**	9.158	**
	(3.669)		(3.029)	
Family_household	0.788			
	(4.942)			
Health_inst	−10.519			
	(14.80)			
Distance	−0.115			
	(0.368)			
Support_family	3.958		6.561	
	(12.06)		(7.616)	
Support_friends	−12.89			
	(7.41)			
Support_neighbours	3.18			
	(10.619)			
Social_support	11.844		6.892	
	(8.466)		(7.272)	
Observations	44		44	
F statistic	1.265		2.707	
R^2^	0.354		0.263	
Adjusted R^2^	0.074		0.166	
Reset test (*p* value)	0.508		0.908	
Jarque Bera test (*p* value)	0.992		0.749	
Breush Pagam test (*p* value)	0.355		0.479	
F statistic for compare both models	0.531			

Note: Estimation method OLS. The standard error of each parameter is in parentheses. The dependent variable is the FES. Model 2 Global includes all the variables under study as the full model, and Model 2 Restricted includes the significant variables. Significance codes: *** *p* < 0.001; ** *p* < 0.01; * *p* < 0.05.

**Table 3 healthcare-13-00791-t003:** Multicollinearity analysis for Model 2 Restricted (presented in Table 2).

Multicollinearity Analysis (VIF)	
Barthel	1.407
Income	1.562
Educ_level	1.646
Support_family	1.348
Social_support	1.139

**Table 4 healthcare-13-00791-t004:** Results of the estimation of Model 3 (Global and Restricted).

Dependent Var	IOFS			
	Model 3 Global		Model 3 Restricted	
(Intercept)	56.01153	*	53.4089	***
	(22.70409)		(7.4166)	
Barthel	−3.22431	**	−2.5889	**
	(1.16119)		(0.8598)	
Sex	3.31467			
	(11.24551)			
Age	−0.03875			
	(0.20792)			
Income	2.91637	*	3.0718	**
	(1.14353)		(0.9848)	
Religion	0.89634			
	(2.93013)			
Educ_level	−2.40455		−2.3833	*
	(1.30602)		(1.1337)	
Family_household	−2.62057		−2.8185	*
	(1.75926)		(1.3828)	
Health_inst	0.49687			
	(5.27006)			
Distance	−0.01911			
	(0.13114)			
Support_family	−3.26156			
	(4.29397)			
Support_friends	3.04333		1.9671	
	(2.63901)		(2.2680)	
Support_Neighbours	−4.54895			
	(3.78021)			
Social_support	−4.83596		−4.7685	
	(3.01383)		(2.6897)	
Observations	44		44	
F statistic	1.85		3.98	
R^2^	0.445		0.399	
Adjusted R^2^	0.205		0.294	
Reset test (*p* value)	0.511		0.429	
Jarque Bera test (*p* value)	0.268		0.241	
Breush Pagam test (*p* value)	0.892		0.3593	
F statistic for compare both models	0.407			

Note: Estimation method OLS. The standard error of each parameter is in parentheses. The dependent variable is IOFS. Model 3 Global includes all the variables under study as the full model, and Model 3 Restricted includes the significant variables. Significance codes: *** *p* < 0.001; ** *p* < 0.01; * *p* < 0.05.

**Table 5 healthcare-13-00791-t005:** Multicollinearity analysis for Model 3 Restricted (presented in Table 4).

Multicollinearity Analysis (VIF)	
Barthel	1.1542
Income	1.640
Educ_level	1.847
Family_household	1.139
Support_friends	1.293
Social_support	1.248

## Data Availability

The datasets used and/or analyzed during the current study are available from the corresponding author upon reasonable request.

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
