# Peer review of "Family Self-Care Pattern in Families with Children with Intellectual Disabilities: A Pilot Study"

_healthcare, 2025, doi:10.3390/healthcare13070791_

Round 1

Reviewer 1 Report

Comments and Suggestions for Authors

The article addresses an important and timely topic concerning patterns of family self-care in families raising children with intellectual disabilities (ID). The aim of the study was to identify factors influencing the development of family self-care and to determine the attributes of four key domains: cognitive, psychosocial, physical and behavioral. This approach addresses a research gap, as the issue of family self-care in this context is still poorly recognized, not only in Portugal, but also in other European countries.

The different parts of the article are structured properly and meet the standards of a scientific article.

The authors used a descriptive-correlational design, which is an apt choice when exploring a new phenomenon. The research sample consisted of 44 families from the Lower Alentejo region, selected using a purposive method. The study used validated tools, including the Barthel Index, Impact on Family Scale and Family Empowerment Scale. Statistical analyses (linear regressions) were performed and supported by Gauss-Markov tests, taking into account the assumptions of normality, homoscedasticity and colinearity.

The results showed that the key factors influencing self-care patterns in families were family income, parent/guardian education level, level of physical dependence of the child on ID, household size and social support network. Interestingly, an inverse correlation was found between income and families' level of empowerment - lower income was associated with a higher sense of empowerment. These findings are in line with previous studies that emphasize the importance of social support and internal resources in families' adaptation to the situation of raising a child with a disability.

The proposed and used literature is correctly selected.  The author aptly discusses the results in relation to the literature. They take into account the diversity of factors affecting the development of self-care and point out the complex psychosocial dynamics of these families. Also appreciated is the extensive reference to theories (Orem, von Bertalanffy, Riegel) that give the study a solid theoretical basis.

Strengths of the developed manuscript include: the push of the study on a comprehensive theoretical model, the use of validated research tools, practical conclusions for family nurses and health care systems.

Also important is the author's inclusion of the perspective of the family as a system.

At the same time, several issues can be pointed out that allow the described research to be treated only in terms of a pilot study.

Key issues include:

1 The small size of the research sample and methodological limitations give the study the stature of a rather pilot study.  The study was conducted with a sample of 44 families from the Lower Alentejo region of Portugal. The author acknowledges that 54 families were originally identified, but only 44 agreed to participate. This sample size is relatively small, which limits the statistical power of the analyses and affects the generalizability of the results. With such a small number of respondents, the risk of errors of the first and second kind in statistical analyses increases. This limits the ability to detect subtle relationships and makes it difficult to apply more sophisticated statistical models. For future surveys, it is worth considering a sample larger than 100 families to increase analytical power and reduce the risk of statistical errors.

2 The selection of families for the study was done through community nurses who recommended specific cases. This is a purposive selection, which potentially introduces a selection bias.

This sampling limits the representativeness of the results and makes it difficult to generalize conclusions to the entire population of families with children with intellectual disabilities - even regionally, let alone nationwide or in other cultures. Moreover, the majority of participants in the study were women (97.7%), which may be due to the approach of the recruiting nurses, who most often interact with mothers as primary caregivers.

3 Although the study is concerned with the family pattern of self-care, in practice, data were collected almost exclusively from mothers (42 of 44 cases). As a result, the analyses are based mainly on their subjective assessment. The omission of other family members, such as fathers, siblings or grandparents, means that the study does not fully reflect the dynamics of the family system. This can lead to a distorted picture of family functioning as a whole. Modeling the pattern of self-care (FSC) can be distorted if the roles, tasks and attitudes of other family members are not taken into account. Efforts should be made to have more family members complete questionnaires, such as both mother and father, or siblings. This will allow for a more complete analysis of family dynamics and a better representation of the FSC pattern. Qualitative methods, such as focus group interviews, can be used to diversify perspectives.

4 As the author points out, the study was conducted in a specific region (Lower Alentejo) that has its own unique cultural and social conditions - including lower income levels and weaker access to public services. The results are not directly transferable to other regions of Portugal or other countries with different demographics, health systems or degree of urbanization.

5 There is a wide variety of diagnoses in the sample (e.g. autism, Down syndrome, cerebral palsy and others). Although the common denominator is intellectual disability, the differences in children's level of functioning are significant. Combining children with different health needs, functional levels and care requirements in one group can distort the results and make it difficult to draw clear conclusions.

Thus, I recommend publication of the reviewed article, however, with a clear indication from the authors that the methodological assumptions carried out give prominence only to pilot studies. In addition, it is worth pointing out the limitations mentioned above as guidelines and recommendations for representative research. 

Author Response

Thank you very much for your review.

Reviewer 2 Report

Comments and Suggestions for Authors

File attached. many parts are extensively written. The reader will not be interested in the write-up. try to condense your methods.

Author Response

Thank you very much for your review.

Round 2

Reviewer 2 Report

Comments and Suggestions for Authors

good revision